

# Geographic disparities and temporal changes of diabetes-related mortality risks in Florida: a retrospective study

Nirmalendu Deb Nath* and Agricola Odoi*

Biomedical and Diagnostic Sciences, University of Tennessee, Knoxville, TN, United States
* These authors contributed equally to this work.

## ABSTRACT

**Background:** Over the last few decades, diabetes-related mortality risks (DRMR) have increased in Florida. Although there is evidence of geographic disparities in pre-diabetes and diabetes prevalence, little is known about disparities of DRMR in Florida. Understanding these disparities is important for guiding control programs and allocating health resources to communities most at need. Therefore, the objective of this study was to investigate geographic disparities and temporal changes of DRMR in Florida.

**Methods:** Retrospective mortality data for deaths that occurred from 2010 to 2019 were obtained from the Florida Department of Health. Tenth International Classification of Disease codes E10–E14 were used to identify diabetes-related deaths. County-level mortality risks were computed and presented as number of deaths per 100,000 persons. Spatial Empirical Bayesian (SEB) smoothing was performed to adjust for spatial autocorrelation and the small number problem. High-risk spatial clusters of DRMR were identified using Tango's flexible spatial scan statistics. Geographic distribution and high-risk mortality clusters were displayed using ArcGIS, whereas seasonal patterns were visually represented in Excel.

**Results:** A total of 54,684 deaths were reported during the study period. There was an increasing temporal trend as well as seasonal patterns in diabetes mortality risks with high risks occurring during the winter. The highest mortality risk (8.1 per 100,000 persons) was recorded during the winter of 2018, while the lowest (6.1 per 100,000 persons) was in the fall of 2010. County-level SEB smoothed mortality risks varied by geographic location, ranging from 12.6 to 81.1 deaths per 100,000 persons. Counties in the northern and central parts of the state tended to have high mortality risks, whereas southern counties consistently showed low mortality risks. Similar to the geographic distribution of DRMR, significant high-risk spatial clusters were also identified in the central and northern parts of Florida.

**Conclusion:** Geographic disparities of DRMR exist in Florida, with high-risk spatial clusters being observed in rural central and northern areas of the state. There is also evidence of both increasing temporal trends and Winter peaks of DRMR. These findings are helpful for guiding allocation of resources to control the disease, reduce disparities, and improve population health.

Corresponding author
Agricola Odoi, aodoi@utk.edu

## INTRODUCTION

Diabetes is a chronic metabolic disease affecting millions of people worldwide and its prevalence has been increasing among the adult population of the United States (US) over the past few decades (*Centers for Disease Control and Prevention, 2003a*). According to the Centers for Disease Control and Prevention, a total of 28.7 million Americans have been diagnosed with diabetes and 8.5 million people are living with undiagnosed diabetes (*Centers for Disease Control and Prevention, 2022b*).

Diabetes is associated with several life-threatening conditions, including chronic kidney disease, cardiovascular disease, stroke, retinopathy, and visual impairment (*Centers for Disease Control and Prevention, 2022a*, *2022b*). As a result, people with diabetes are at a higher risk of death compared to those without the condition. As of 2021, the Diabetes Related Mortality Risk (DRMR) in the US was 31.1 per 100,000 persons, and the disease is reported to be the 8th leading cause of death in the country (*Centers for Disease Control and Prevention, 2023b*; *Xu et al., 2022*). The condition imposes a significant economic burden, as the average medical expenses of patients with diabetes are 2.3 times higher than for those who do not have the disease (*Florida Department of Health, 2017*). In 2017, Florida's total diabetes related costs was approximately $25 billion, with $19.3 billion being direct costs and another $5.5 billion being indirect costs (*Florida Department of Health, 2017*).

There is evidence of geographic disparities of diabetes in the US. The diabetes belt, which includes fifteen states of the Southeast US (including Florida), has higher diabetes prevalence (≥11.0%) compared to the nation's average (8.5%) (*Barker et al., 2011*). Every year an estimated 148,613 people are diagnosed with diabetes in Florida (*Florida Department of Health, 2017*), and as of 2021, the DRMR was 24.8 per 100,000 persons (*Centers for Disease Control and Prevention, 2022c*). There has been a significant upward trend in DRMR in the US over the last few decades (*US Department of Health and Human Services, 2020*) and these temporal changes show evidence of regional disparities (*US Department of Health and Human Services, 2020*) as certain states and communities have experienced higher mortality risks than others. Moreover, the risk of diabetes increases with age and Florida has the 2nd largest proportion of seniors (*i.e.* individuals >65 years old). Addressing geographic disparities of DRMR is important in providing useful information to guide efforts to reduce disparities and improve population health. Therefore, the objective of this study was to identify geographic disparities and temporal changes of DRMR in Florida.

## METHODS

### Ethics approval

This study was approved by the University of Tennessee, Knoxville Institutional Review Board (Number: UTK IRB-23-07809-XM).

## Study area

This retrospective study was conducted in the state of Florida, which comprises 67 counties (Fig. 1), some of which lie within the diabetes belt (*Barker et al., 2011*). Geographically, the state is located between 27° 66′ N and 81° 52′ W and spans 65,758 square miles, ranking 22nd by area among the 50 states of the US. As of 2020, 21.5 million people live in Florida, of whom 50.8% are female and 49.2% are male (*United States Census Bureau, 2023a*). The age distribution among the adult population is as follows: 24% are 18–34 years old, 26% are 35–49 years old, 25% are 50–64 years old, and 22% ≥ 65 years old. The overall racial composition of Florida residents is 76.9% White, 17% Black, and 6.1% all other races (*United States Census Bureau, 2023a*). There are 71.2% non-Hispanic, and 28.73% Hispanic population. Miami-Dade County, which is located in the Southeastern part of the state, is the most populous with 2.6 million people, whereas Liberty County is the least populated with 7,987 residents (*United States Census Bureau, 2023a*). Counties with population densities of ≤100 persons per square mile are classified as rural county, while those with higher population densities are classified as urban county (*Florida Department of Health, 2023a*). Based on this classification, there are 30 rural and 37 urban counties in Florida (Fig. 1).

## Data source and management

The 2010–2019 individual-level death data were obtained from the *Florida Department of Health (2023b)*. The cause of death was recorded using the 10th revision of the International Classification of Disease (*World Health Organization, 2023*), and the codes E10–E14 were used to identify diabetes-related deaths (*World Health Organization, 2023*). No differentiation was made between Type 1 and Type 2 diabetes. The number of diabetes-related deaths were aggregated at the county level using R statistical software version 4.2.2 (*R Core Team, 2023*). County-level DRMR were then calculated and expressed as number of deaths per 100,000 persons. Population estimates for 2010 to 2019 were obtained from the American Community Survey and used as the denominators to calculate the county-level DRMR from 2010 to 2019 time periods (*United States Census Bureau, 2022*). Cartographic boundary files were downloaded from the United States Census Bureau TIGER Geodatabase (*United States Census Bureau, 2023b*) and used for the spatial displays at the county level. GeoDa software version 1.14 (*Anselin, 2023*) was used to compute the county-level Spatial Empirical Bayesian (SEB) smoothed DRMR to adjust for spatial autocorrelation and small number of cases in some counties (*Bernardinelli & Montomoli, 1992*; *Khan et al., 2023*; *Haddow & Odoi, 2009*).

## Detection of spatial clusters

Evidence of spatial autocorrelation of DRMR was assessed using global Moran's I statistic specifying 1$^{st}$ order queen weights and implemented in GeoDa (*Anselin, 2023*). Tango's Flexible Spatial Scan Statistics (FSSS) was used in FleXScan software version 3.1.2 (*Takahashi, Yokoyama & Tango, 2010*) to identify statistically significant irregularly shaped and circular high risk spatial clusters of DRMR (*Tango & Takahashi, 2005*). The maximum cluster size of 15 counties was specified as spatial scanning window to avoid

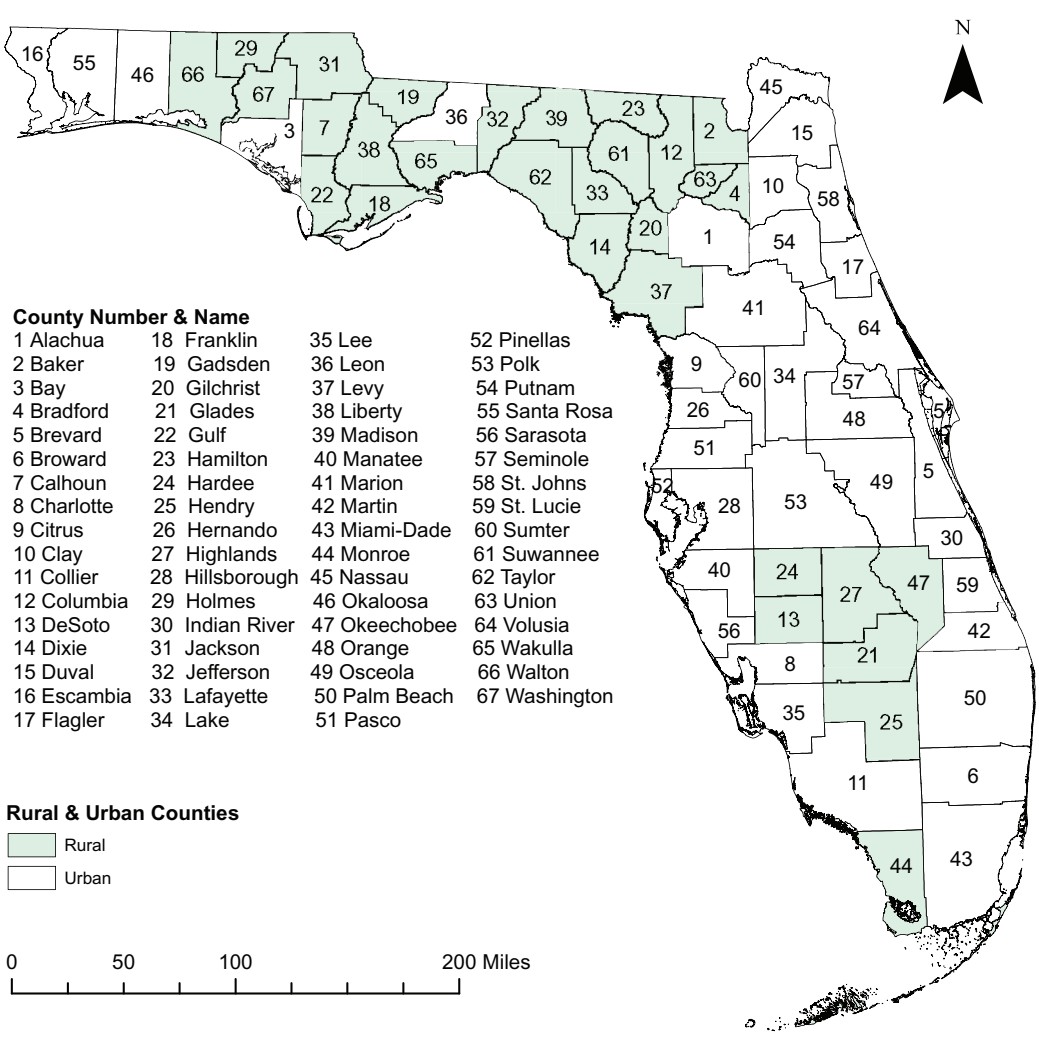

**County Number & Name**

| | | | |
|---|---|---|---|
| 1 Alachua | 18 Franklin | 35 Lee | 52 Pinellas |
| 2 Baker | 19 Gadsden | 36 Leon | 53 Polk |
| 3 Bay | 20 Gilchrist | 37 Levy | 54 Putnam |
| 4 Bradford | 21 Glades | 38 Liberty | 55 Santa Rosa |
| 5 Brevard | 22 Gulf | 39 Madison | 56 Sarasota |
| 6 Broward | 23 Hamilton | 40 Manatee | 57 Seminole |
| 7 Calhoun | 24 Hardee | 41 Marion | 58 St. Johns |
| 8 Charlotte | 25 Hendry | 42 Martin | 59 St. Lucie |
| 9 Citrus | 26 Hernando | 43 Miami-Dade | 60 Sumter |
| 10 Clay | 27 Highlands | 44 Monroe | 61 Suwannee |
| 11 Collier | 28 Hillsborough | 45 Nassau | 62 Taylor |
| 12 Columbia | 29 Holmes | 46 Okaloosa | 63 Union |
| 13 DeSoto | 30 Indian River | 47 Okeechobee | 64 Volusia |
| 14 Dixie | 31 Jackson | 48 Orange | 65 Wakulla |
| 15 Duval | 32 Jefferson | 49 Osceola | 66 Walton |
| 16 Escambia | 33 Lafayette | 50 Palm Beach | 67 Washington |
| 17 Flagler | 34 Lake | 51 Pasco | |

**Rural & Urban Counties**

Rural
Urban

0      50      100      200 Miles

**Figure 1 Study area showing the geographic distribution of rural and urban counties in Florida, USA.** Base map source credit: United States Census Bureau, https://www.census.gov/geographies/mapping-files/time-series/geo/tiger-line-file.2019.html. 

the identification of excessively large clusters. Poisson probability model with a restricted log likelihood ratio (LLR) was utilized (*Tango & Takahashi, 2012*). Nine hundred and ninety-nine Monte Carlo replications and a critical *p*-value of 0.05 were used for significance testing to identify statistically significant high-risk clusters. Potential clusters were then ranked based on their restricted LLR. The cluster with highest restricted LLR value was designated as the primary cluster and the rest were considered secondary clusters. Only high-risk clusters with relative risk of ≥1.20 were reported in this study in order to avoid reporting low risk clusters. To assess if the spatial distribution of the clusters changed between the beginning of the study period and the end, two cluster analyses were performed: one for the first 2 years (2010–2011) and another for the last 2 years (2018–2019) of the study period. The spatial distribution of the clusters were then compared visually.

### Identification of temporal changes

The descriptive statistics of DRMR were calculated using R statistical software version 4.2.2 (*R Core Team, 2023*) implemented in RStudio (*Rstudio Team, 2023*). The number of diabetes associated deaths were aggregated by season (winter: December, January, February; spring: March, April, May; summer: June, July, August; and fall: September, October, November) to identify the temporal changes of DRMR for the period of 2010 to 2019. Seasonal DRMR were computed and expressed as number of deaths per 100,000 persons. The temporal changes in DRMR over the 10-year study period were displayed graphically in Microsoft Excel (Microsoft, Redmond, WA, USA).

### Cartographic displays

The cartographic displays were performed using ArcGIS version 10.8.1 (*Environmental Systems Research Institute, 2023*). County-level choropleth maps were used to visualize the distribution of both smoothed and unsmoothed DRMR using Jenk's optimization classifications scheme (*Jenks, 1967*). Choropleth maps, using Jenk's optimization classification scheme, were generated for each of the 10 years of the study period. Identified high-risk spatial clusters of DRMR were also displayed in ArcGIS.

## RESULTS

### Geographic distribution of DRMR

A total of 54,684 diabetes-related deaths were reported in Florida during the study period. The spatial patterns of the SEB smoothed maps were more apparent than those of the unsmoothed maps (Figs. 2 and 3). The county-level SEB DRMR varied geographically and ranged from 12.6 to 81.1 deaths per 100,000 persons across the state. The lowest mortality risk was observed in Leon County (12.6 deaths per 100,000 persons in 2015), while the highest risk was in Taylor County (81.1 per 100,000 persons in 2018). The northern and central counties of Florida tended to have high mortality risks during 2010–2011 and 2016–2019 time periods (Fig. 3). High DRMRs were observed predominantly in rural counties (Fig. 4). Specifically, the highest mortality risk (58.7 deaths per 100,000 persons) was recorded in northern rural Taylor County, while urban Leon County had the lowest (18.6 deaths per 100,000 persons) mortality risks. The rural counties neighboring Taylor County (*i.e.* Lafayette, Madison, Suwannee, Hamilton, Columbia, and Highlands counties) had similarly high mortality risks (Figs. 1 and 4). Although a few southern urban counties (including Desoto, Hardee, Highlands, and Glades counties) had high mortality risks, most of the the urban counties in the southern parts of the state had low mortality risks (Figs. 1 and 4).

### Purely spatial clusters of DRMR

There was evidence of global spatial autocorrelarion of DRMR (Morans I = 0.430; $p < 0.001$). Significant high-risk spatial clusters were detected in the central and northern parts of the state (Fig. 5). Six and seven clusters were identified during the 2010–2011 and 2018–2019 time periods, respectively (Table 1). More counties were involved in the spatial

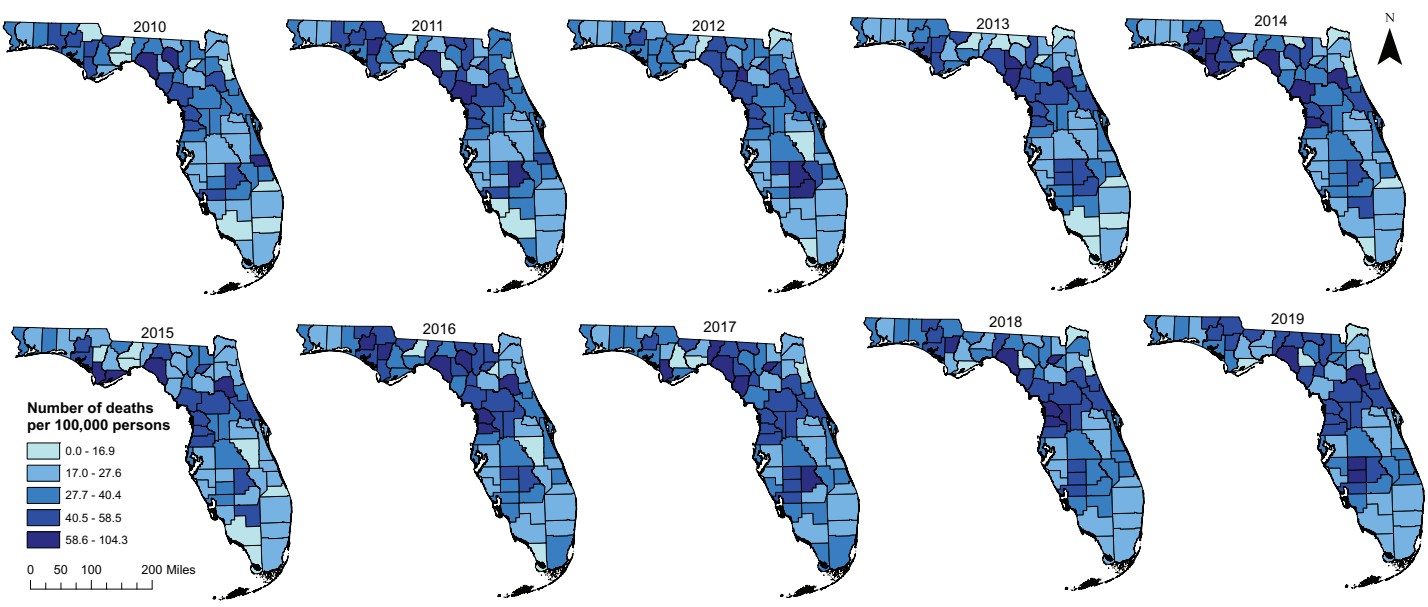

**Figure 2 Distribution of unsmoothed diabetes-related mortality risks in Florida, 2010 to 2019.** Base map source credit: United States Census Bureau, https://www.census.gov/geographies/mapping-files/time-series/geo/tiger-line-file.2019.html.

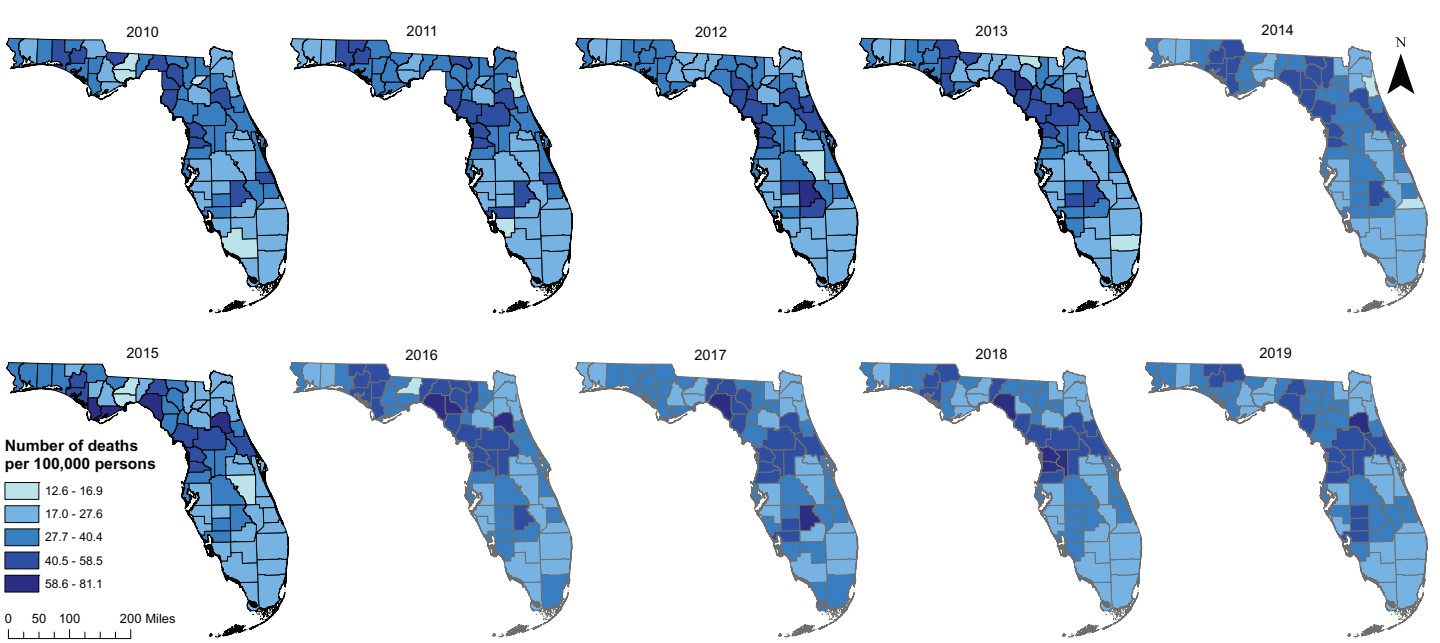

**Figure 3 Distribution of spatial empirical bayes smoothed diabetes-related mortality risks in Florida, 2010 to 2019.** Base map source credit: United States Census Bureau, https://www.census.gov/geographies/mapping-files/time-series/geo/tiger-line-file.2019.html.

clusters during the 2018–2019 time period (29 counties) than the 2010–2011 time period (28 counties). In 2010–2011, most of the high-risk clusters were identified in the northern parts of the state; however, by 2018–2019, a subtle shift had taken place, as the high-risk clusters moved from the northern parts to the central areas of Florida. Most of the counties involved in the high-risk clusters were located in the rural areas, although some urban

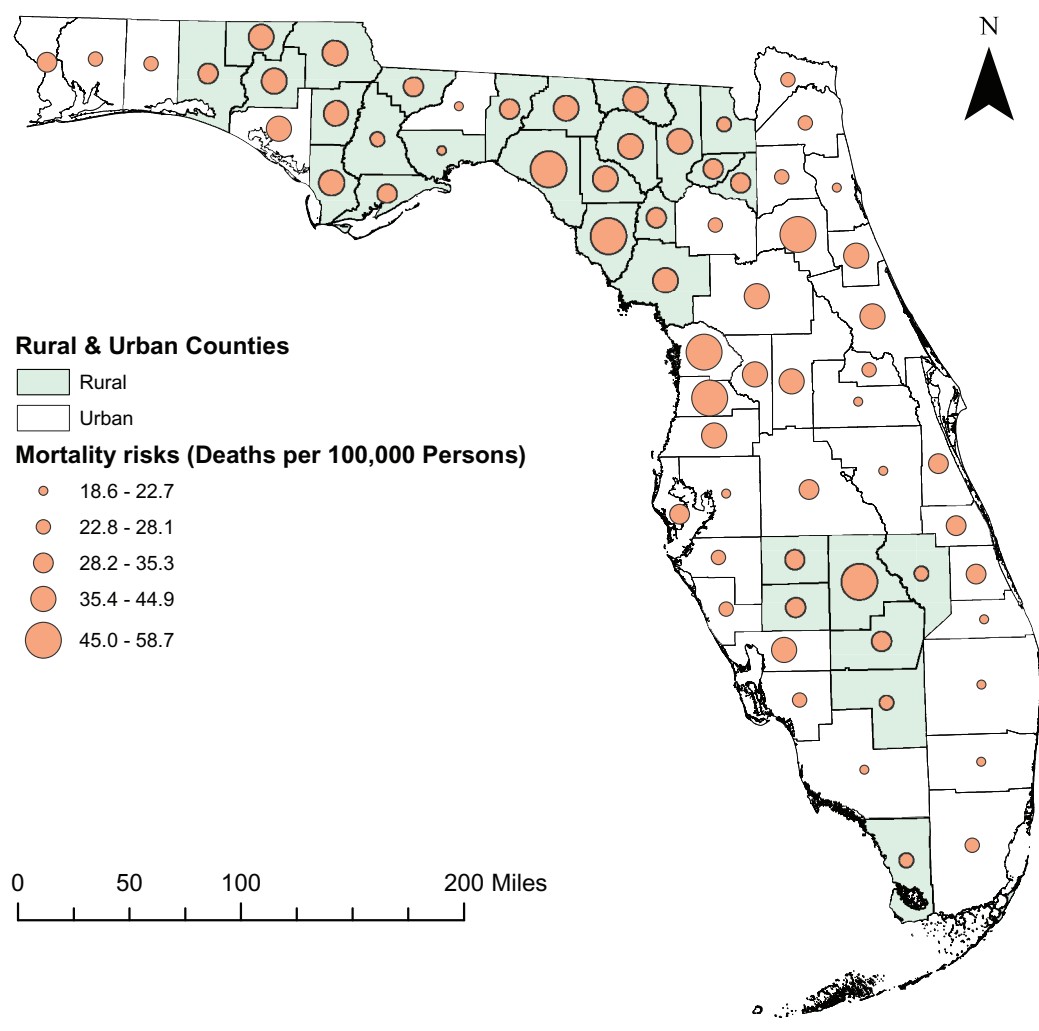

N

**Rural & Urban Counties**

Rural

Urban

**Mortality risks (Deaths per 100,000 Persons)**

○  18.6 - 22.7

○  22.8 - 28.1

○  28.2 - 35.3

○  35.4 - 44.9

○  45.0 - 58.7

0     50     100              200 Miles

**Figure 4  Distribution of diabetes-related mortality risks in rural and urban counties of Florida, 2010 to 2019.** Base map source credit: United States Census Bureau, https://www.census.gov/geographies/mapping-files/time-series/geo/tiger-line-file.2019.html.     

counties (Marion, Sumter, Citrus, Hernando, and Pasco) were also part of high-risk clusters.

### Temporal changes of DRMR

There was an increasing temporal trend and seasonal patterns in DRMR from 2010 to 2019 (Fig. 6). The risks were consistently high during the winter season and low during the fall season (Fig. 6). The lowest DRMR was recorded during the fall season of 2010 (6.08 deaths per 100,000 persons) while the highest was recorded during the winter of 2018 which had a risk of 8.13 deaths per 100,000 persons.

## DISCUSSION

This study investigated geographic disparities and temporal patterns of county-level DRMR in Florida. The findings of this study will be useful for guiding evidence-based

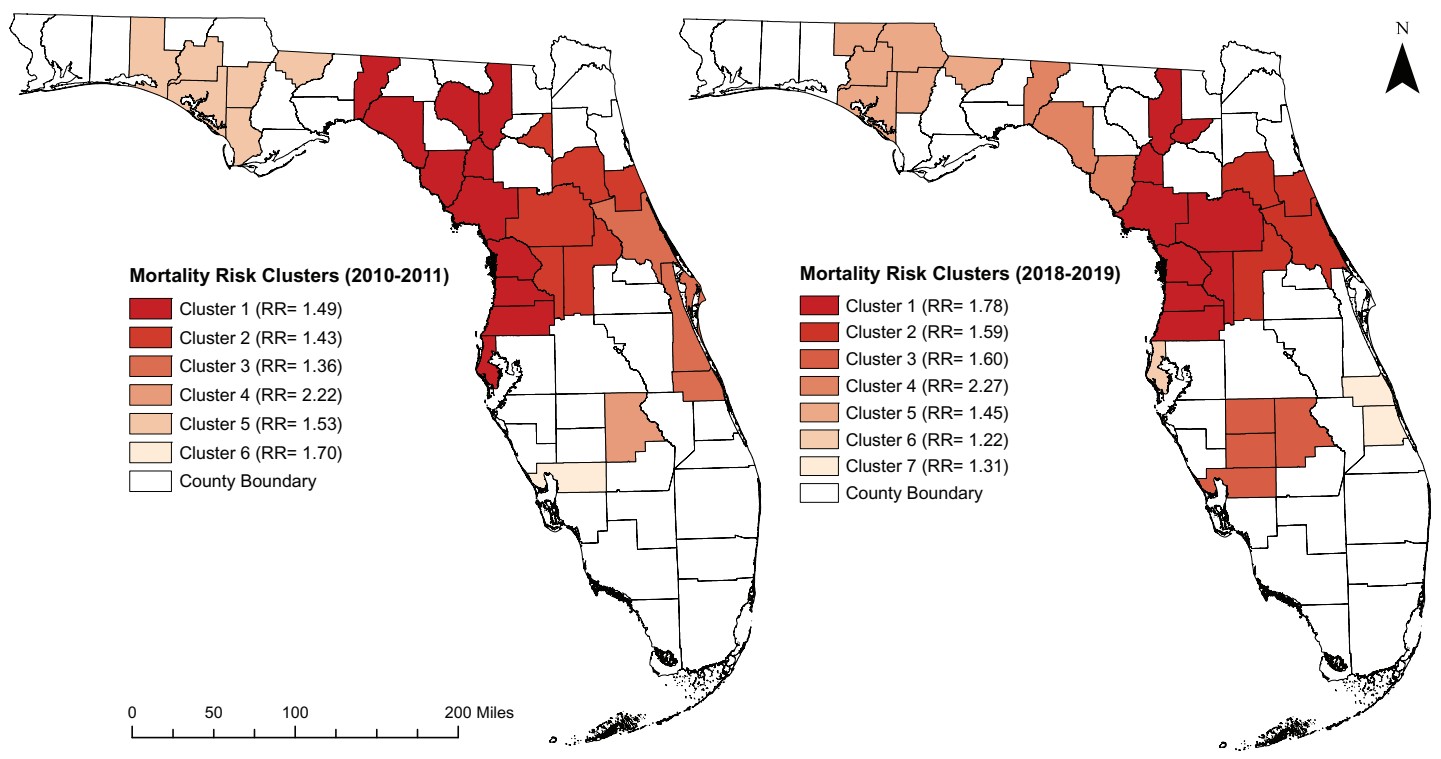

**Figure 5** **High risk spatial clusters of DRMR in Florida during the time period of 2010–2011 and 2018–2019.** Base map source credit: United States Census Bureau, https://www.census.gov/geographies/mapping-files/time-series/geo/tiger-line-file.2019.html.

health planning and resource allocation to reduce disparities and improve diabetes health outcomes by targeting high risk communities with control programs.

## Geographic distribution and clustering of DRMR

There is evidence of geographic disparities of DRMR in Florida, with most of the high-risk counties being located in the rural northern and central parts of the state. Some previous studies also consistently reported that diabetes mortality remains a significant cause of concern in rural America, especially in the southern rural parts of the US (*Callaghan et al., 2020*; *Brown-Guion et al., 2013*; *O'Brien & Denham, 2008*). These rural areas experienced much higher DRMR compared to their urban counterparts (*Callaghan et al., 2020*), indicating a significant disparity between rural and urban areas. The lack of healthcare facilities in rural parts of Florida might be a reason for these disparities (*Rural Health for Florida, 2023*). Additionally, insufficient public transportation and long distances to healthcare facilities may also hinder the accessibility of healthcare services (*O'Brien & Denham, 2008*; *Agency for Healthcare Research and Quality, 2023*). Rural communities face challenges in attending regular health checkups, leading to lower detection rates of potential health problems, including diabetes, and subsequently increase the risks of diabetes-related deaths.

Significant high-risk spatial clusters of DRMR exist in the northern and central parts of the state. The result is consistent with the previous findings, which reported that some

**Table 1 Purely spatial clusters of high DRMR identified in Florida from 2010–2011 and 2018–2019.**

| Period | Cluster | Population | Observed cases | Expected cases | No. of counties | RR* | p-value |
|---|---|---|---|---|---|---|---|
| 2010–2011 | Cluster 1 | 1,904,985 | 1,053 | 1,008 | 10 | 1.49 | 0.001 |
| | Cluster 2 | 926,600 | 697 | 486 | 6 | 1.43 | 0.001 |
| | Cluster 3 | 1,180,052 | 843 | 619 | 3 | 1.36 | 0.001 |
| | Cluster 4 | 987,67 | 115 | 51 | 1 | 2.22 | 0.001 |
| | Cluster 5 | 328,536 | 263 | 172 | 6 | 1.53 | 0.001 |
| | Cluster 6 | 161,115 | 144 | 84 | 1 | 1.70 | 0.001 |
| 2018–2019 | Cluster 1 | 1,180,597 | 1,535 | 860 | 7 | 1.78 | 0.001 |
| | Cluster 2 | 1,077,748 | 982 | 617 | 4 | 1.59 | 0.001 |
| | Cluster 3 | 348,718 | 320 | 199 | 4 | 1.60 | 0.001 |
| | Cluster 4 | 54,010 | 70 | 30 | 3 | 2.27 | 0.001 |
| | Cluster 5 | 318,480 | 282 | 194 | 5 | 1.45 | 0.001 |
| | Cluster 6 | 979,558 | 687 | 561 | 1 | 1.22 | 0.001 |
| | Cluster 7 | 464,381 | 348 | 266 | 2 | 1.31 | 0.001 |

**Note:**
* Relative Risk.

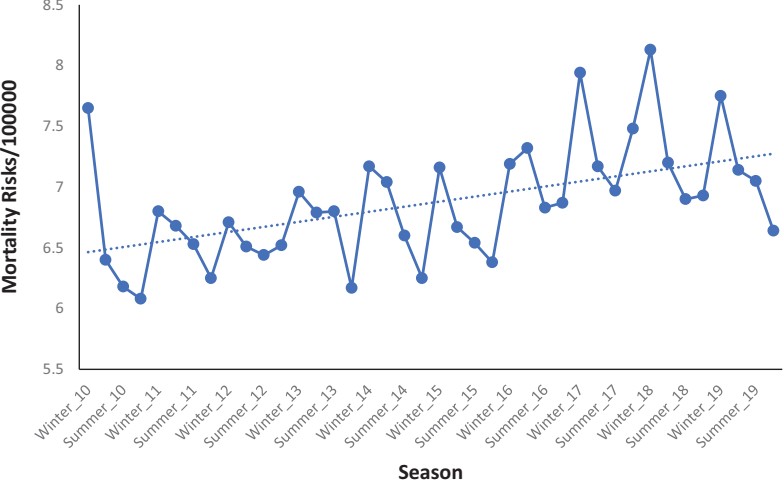

**Figure 6 Seasonal patterns of diabetes related mortality risks in Florida, 2010–2019.**

counties in the north regions of Florida are located within the diabetes belt and have excess diabetes prevalence and diabetes-related complications (*Barker et al., 2011*; *Ford et al., 2012*). Geographic disparities in the Diabetes Self-Management Education program, which aims to educate diabetic patients on disease management, might be a reason behind these observed results. Some previous studies reported that the implementation of Diabetes Self-Management Education program remains low in northern counties of Florida despite these regions having high prevalence (*Paul et al., 2018*; *Khan et al., 2021*). As a result, residents of the counties of north Florida are not aware of diabetes and diabetes-associated complications, hence increasing the risks of death.

Differences in socioeconomic conditions of the communities might be another reason for the observed cluster patterns. Most of the counties with below median household income are located in the northern and central regions of Florida (*Florida Department of Health, 2023c*). This disparity in income levels may result in limited access to essential health insurance, preventive care, proper nutrition, and physical activity (*Loftus et al., 2017*), directly or indirectly influencing diabetes and associated complications. Studies report that adults with diabetes living in societies of low-income brackets are at a two-fold higher risk of death compared with their affluent family counterparts (*Saydah & Lochner, 2010*; *Mackenbach, 2012*; *Dalsgaard et al., 2015*). It is also possible that geographic differences in comorbidities may play a role in the observed DRMR clustering. Previous research indicated that the northern parts of Florida exhibit significantly higher rates of comorbidities, including hypertension, heart disease, and kidney disease, compared to other regions within the state (*Smith et al., 2018*; *Odoi et al., 2019*). Comorbidities in individuals with diabetes can further increase the risk of mortality through their adverse effects on blood pressure, myocardial infarction, cancer risks, kidney function, and blood sugar control. Some previous studies already reported that cardiovascular disease is a strong predictor of diabetes-related mortality, and it was three to four times higher in patients with diabetes than those without diabetes (*Khaw et al., 2004*; *Rawshani et al., 2017*; *Lu et al., 2021*).

## Temporal pattern of DRMR

The observed high mortality risks during the winter months, followed by a decline from summer to fall and a subsequent increase in the winter season throughout the study period, were comparable with reports from other previous studies (*Zhang et al., 2021*; *Tseng et al., 2005*). A recent study conducted by *Zhang et al. (2021)* reported that the mean level of fasting blood glucose is significantly higher during the winter than summer, which may contribute to an increased risk of diabetes-related complications and death. Additionally, hemoglobin A1c (glycosylated hemoglobin or A1c) which is an indicator of the past 90 days' average blood glucose level, is also associated with microvascular and macrovascular risk in diabetes patients (*Teitelbaum et al., 1990*), and fluctuation of A1c level can increase risk of death. *Tseng et al. (2005)* recently reported that A1c level was higher during the winter season and lower in summer with a difference of 0.22 A1c unit. A number of factors such as physiological, dietary, body mass index, and physical activity changes in the winter might be possible explanations for the observed seasonal pattern of DRMR. Physical inactivity and sedentary behavior have been reported to increase in winter months (*Cepeda et al., 2018*; *Pivarnik, Reeves & Rafferty, 2003*), which also increases high blood glucose as well as diabetes-related deaths.

## Strengths and Limitations

This study used a rigorous spatial epidemiological approach to investigate the geographic distribution and spatial clusters of DRMR in Florida. Tango's FSSS is a robust method of identifying spatial clusters as it does not involve multiple comparisons, and it identifies both circular and irregularly shaped clusters. Moreover, this is the first study that has

investigated geographic disparities of DRMR in Florida. However, this study has some limitations. Spatial patterns identified at the county levels may be different from those at lower spatial scales of analysis such as census tracts. Unfortunately, due to the small number of cases of diabetes deaths at the lower spatial scales, these analyses could not be conducted at these lower spatial sclaes. Tango's FSSS has low power for detecting circular clusters. Additionally, this study used surveillance data which might have some limitations in terms of case attainment and reporting. Finally, this study only investigated geographic disparities of DRMR at the county level; therefore, it was not possible to assess/identify lower-level disparities.

## CONCLUSIONS

Geographic disparities in diabetes-related mortality risks exist in Florida, with notable high-risk clusters in the rural central and northern regions of the state. There was also evidence of increasing temporal trend and seasonal patterns of DRMR with the highest mortality risks being observed during the winter season. The study findings are useful for guiding health resource allocation targeting high risk communities so as to reduce health disparities in Florida. Future research will investigate predictors of the identified disparities.

## ACKNOWLEDGEMENTS

The authors are grateful to the Florida Department of Health for providing the data.

### Funding

The authors received no funding for this work.

### Competing Interests

Agricola Odoi is an Academic Editor for PeerJ.

### Author Contributions

- Nirmalendu Deb Nath conceived and designed the experiments, performed the experiments, analyzed the data, prepared figures and/or tables, authored or reviewed drafts of the article, and approved the final draft.
- Agricola Odoi conceived and designed the experiments, performed the experiments, authored or reviewed drafts of the article, and approved the final draft.

### Human Ethics

The following information was supplied relating to ethical approvals (*i.e.*, approving body and any reference numbers):

The University of Tennessee Institutional Review Board (IRB) granted ethical approval to conduct the study (IRB Approval Number: UTK IRB-23-07809-XM).

## Data Availability

The data is available in the Supplemental File.

## Supplemental Information

Supplemental information for this article can be found online at http://dx.doi.org/10.7717/peerj.17408#supplemental-information.

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
