# Peer review of "Geographic disparities and temporal changes of diabetes-related mortality risks in Florida: a retrospective study"

_PeerJ, doi:10.7717/peerj.17408_

## Round 0.1 · original submission · Major Revisions

Four reviewers gave reasonable comments and their questions should be answered appropriately.

Reviewer 1 ·

Basic reporting

Clear English was used, and sufficient Background was written.
Raw data were also shared.

Experimental design

Research question well defined, and it was dealt with in the results.

Validity of the findings

All underlying data have been provided. Conclusions were stated based on the results.

Additional comments

The manuscript seems to be well-written. I have some minor comments.

1. Title; I think that an ecological study is a kind of studies that investigate an association between disease and predictors using regional data. However, your study did not investigate an association with predictors, and it might not be called as an ecological study.
2. Introduction; It is not certain why the authors focused on data of Florida. If it is important to investigate geographical difference in diabetes mortality in the U.S., it is meaningful to investigate it using all the states in the U.S..
3. Methods; What R package was used for deriving SEB smoothed DRMR?
4. Results; How about comparing DRMRs in urban and rural areas or showing DRMRs by urbanization level of counties?
5. Discussion; It is mentioned that socioeconomic factors might be a reason for the geographic difference. I think that it can be easily verified in this study by showing an association between income level and DRMR.

Reviewer 2 ·

Basic reporting

Figures are relevant to the content of the article, but their resolution is insufficient, and appropriately described and labeled.

Experimental design

No comment.

Validity of the findings

The data on which the conclusions are based must be checked by a statistician too, although in my opinion, the data are robust, statistically sound, and controlled.

Additional comments

This is a well-done survey, although I think you can make it summarized.

Reviewer 3 ·

Basic reporting

1. It is confusing for the reader to understand the abbreviations (e.g., CD, CE, Ch., etc.) created by the authors throughout the manuscript. I recommend the authors spell out all those terms.
2. The authors justified the rational for using "Spatial Empirical Bayesian (SEB) smoothed DRMR to adjust for spatial autocorrelation and small number of cases in some counties". However, no mention of the existence of spatial autocorrelation in the Results section.

Experimental design

This study could benefit if including analysis of the associated factors of diabetes-related motility risks.

Validity of the findings

Another major concern is the coarse resolution of the data that have been aggregated to counties and there are only 67 counties in the state of Florida. The sample size could have biased the results of local spatial clustering. This limitation should be discussed by the authors at the end of the manuscript.

Reviewer 4 ·

Basic reporting

The manuscript conforms to the journal's guidelines by implementing measures for data deidentification and maintaining ethical standards. The concerns raised in the previous review have been appropriately addressed. Additionally, comprehensive examinations of the figures and tables in the manuscript have been conducted. In general, I commend the authors for their diligent work in presenting a well-structured research layout. The figures are relevant and appropriately labeled, contributing to a clearer understanding of the content.

Experimental design

However, the manuscript still requires minor revisions.
The Data source and management section could benefit from a comparison of diabetes patients in the US from a source such as the National Inpatient Sample Database. It could really help to put into perspective the cases in Florida compared to the US.
Line 133-134: Any particular reason other than the large cluster restriction to choose 15?
Line 140-141: The distribution of RR could help readers understand the reason to choose 1.20 as a threshold to report clusters.

Validity of the findings

Line 172-173: Are Taylor county and Leon county considered rural/urban counties? Di authors investigate the reason why these two counties are ranked lowest and highest in MR? I think a bit more of socio-economic analysis along with access to healthcare of the counties will help make stronger inferences about the results.
Discussion is robust, however, did the authors encounter any challenges with missing data points? If yes, how did the statistical analysis proceed with missing data points? In the data file, Hamilton county in 2013 has 0 mortality risk. What could be the reason for this? Is this a missing data point?

---

## Round 0.2 · accepted · Accept

The comments raised by the reviewers were appropriately addressed.

Reviewer 1 ·

Basic reporting

No comment.

Experimental design

No comment.

Validity of the findings

No comment.

Additional comments

Thank you for the revision. I confirmed it.

Reviewer 3 ·

Basic reporting

The revised version of the manuscript is in much better shape.

Experimental design

The experiment design is well justified and solid.

Validity of the findings

The findings are insightful to the existing literature and public health policies.